# Automatic Segmentation and Quantitative Assessment of Stroke Lesions on MR Images

**DOI:** 10.3390/diagnostics12092055

**Published:** 2022-08-24

**Authors:** Khushboo Verma, Satwant Kumar, David Paydarfar

**Affiliations:** 1Department of Neurology, Dell Medical School, University of Texas, Austin, TX 78712, USA; 2Center for Perceptual Systems, University of Texas, Austin, TX 78712, USA

**Keywords:** stroke, automatic segmentation, stroke, deep neural networks, 3D-UNet, lesion studies, vascular neurology, precision medicine

## Abstract

Lesion studies are crucial in establishing brain-behavior relationships, and accurately segmenting the lesion represents the first step in achieving this. Manual lesion segmentation is the gold standard for chronic strokes. However, it is labor-intensive, subject to bias, and limits sample size. Therefore, our objective is to develop an automatic segmentation algorithm for chronic stroke lesions on T1-weighted MR images. Methods: To train our model, we utilized an open-source dataset: ATLAS v2.0 (Anatomical Tracings of Lesions After Stroke). We partitioned the dataset of 655 T1 images with manual segmentation labels into five subsets and performed a 5-fold cross-validation to avoid overfitting of the model. We used a deep neural network (DNN) architecture for model training. Results: To evaluate the model performance, we used three metrics that pertain to diverse aspects of volumetric segmentation, including shape, location, and size. The Dice similarity coefficient (DSC) compares the spatial overlap between manual and machine segmentation. The average DSC was 0.65 (0.61–0.67; 95% bootstrapped CI). Average symmetric surface distance (ASSD) measures contour distances between the two segmentations. ASSD between manual and automatic segmentation was 12 mm. Finally, we compared the total lesion volumes and the Pearson correlation coefficient (ρ) between the manual and automatically segmented lesion volumes, which was 0.97 (*p*-value < 0.001). Conclusions: We present the first automated segmentation model trained on a large multicentric dataset. This model will enable automated on-demand processing of MRI scans and quantitative chronic stroke lesion assessment.

## 1. Introduction

Over the past century, human lesion studies have provided key insights into the relationship of brain areas to behavior. Lesion studies are highly valuable in cognitive neuroscience as lesions such as stroke create a dissociation of function. This dissociation provides causal evidence of a brain area for the function [1].

In this quest, automatic lesion segmentation has emerged as a crucial tool [2]. Automatic image segmentation refers to assigning non-overlapping boundaries in an image to regions dissimilar in core features such as intensity or texture [3]. There are numerous automatic segmentation algorithms for acute stroke; however, in comparison, there is a paucity of algorithms for chronic stroke lesions [2]. The reasons for this are multifold. First, acute stroke requires time-sensitive and precise clinical decision-making. Neuroimaging plays a pivotal role in this decision-making; therefore, there are concerted efforts to develop automatic segmentation models for acute stroke [4,5,6]. Second, the lesions in acute and chronic stroke are assessed with distinct MR imaging sequences, and segmentation methods developed for acute stroke imaging are not directly applicable to chronic stroke imaging [7]. In acute stroke, diffusion, perfusion, and susceptibility-weighted images offer the most clinically relevant information [8]. In contrast, in chronic stroke, T1-weighted images are preferred as they offer a high spatial resolution, making them suitable for studying the structure-function relationship [7]. Third, imaging in acute stroke is mandatory. Therefore, large neuroimaging datasets exist, which enable automatic model training and testing for acute stroke [9]. In comparison, large chronic stroke imaging datasets are rare [10].

Due to these reasons, currently, manual segmentation is the gold standard for chronic stroke lesions. However, it is labor-intensive, time-consuming, prone to inter-observer variability and biases [9]. All these factors ultimately limit the sample size. Chronic stroke automatic segmentation models are required to overcome these limitations.

Recent reports have described an improvement in the performance of stroke lesion segmentation algorithms when using novel methods, for instance, deep learning and convolutional neural networks [9]. However, most of these algorithms were developed using a small dataset such as ATLAS v1.2 and limited training size, hence, running the risk of overfitting and lacking generalization [9]. It was also reported by Liew et al. that, out of 17 different methods published using ATLAS v1.2, 12 of them did not report the code as publicly available, thereby limiting the utility of the method to the scientific community [9]. As a result, we developed a deep learning model based on the recently released ATLAS v2.0 dataset (Anatomical Tracings of Lesions After Stroke, version 2 [9], accessed in 16 February 2022) and released our code and trained models for the public.

This study aims to develop a fully automatic algorithm to segment chronic stroke lesions on T1-weighted MR images. We used a deep neural network architecture (DNN) to develop our model. Among the machine learning algorithms, DNN is a class that has become well-established in computer vision tasks such as automatic segmentation [11]. Recent studies have reported that DNNs, specifically 3D-UNet, effectively segment stroke lesions [7,12,13,14,15,16,17]. As a variation of 2D-UNet, 3D-UNet takes three-dimensional volumes as inputs [18]. The encoder and decoder structures are identical to those in the 2D-UNet. The encoder, however, extracts the features using 3D convolution followed by 3D max-pooling, and the decoder reconstructs the annotated images using 3D upsampling [18]. Therefore, we chose 3D-UNet as it is suitable for volumetric segmentation of medical imaging [19].

The rest of the paper follows the following structure. We begin by describing how the model was developed. A qualitative and quantitative evaluation of the model will follow in the next section. Finally, we conclude by discussing the application of our developed model in both basic and clinical research.

Overall, we found that DNN enabled objectives and automated assessments of chronic lesions secondary to stroke in vascular neurology at high throughput and could ultimately serve to improve clinical decision-making. Our model, in conjunction with voxel-based lesion-symptom mapping, will facilitate the study of brain-behavior relationships in chronic stroke.

## 2. Materials and Methods

### 2.1. Dataset

We employed an open-source dataset: ATLAS v2.0 (Anatomical Tracings of Lesions After Stroke, version 2). This dataset has 655 T1-weighted MR images in a training set assembled from worldwide multicentric cohort sites as a part of the ENIGMA Stroke Recovery Group study [10]. Each site obtained ethical approval, and was conducted in accordance with the Declaration of Helsinki, 1964. In addition, the central image receiving site obtained ethical approval for the reception and dissemination of the deidentified MR images.

### 2.2. Manual Tracing

Manual tracers followed standard operation protocol to trace lesion labels on the raw T1-weighted images. This was done using ITK-SNAP software (version 3.8.0) [20,21]. The lesion masks were assessed by two raters separately and edited based on a standardized protocol. The link to the standardized protocol is provided in the supplementary materials.

### 2.3. Preprocessing and Dataset Partitioning

Preprocessing included intensity normalization, registration to a standardized template (MNI-152), and defacing using the “mri_deface” tool from FreeSurfer (v1.22) (https://surfer.nmr.mgh.harvard.edu/fswiki/mri_deface). Automated brain extraction using the HD-BET algorithm was performed on all the MR images [22]. HD-BET is a rigorously validated deep learning algorithm that uses artificial neural networks. During our initial testing, we found that the brain extraction (skull removal) process reduces the training time and improves the model performance. For training, the dataset was partitioned into unique five-folds containing 20% of the data in each fold.

### 2.4. Lesion Characteristics

Lesion characteristics are depicted in Table 1. 59.9% of subjects had a single lesion, and 38.1% had multiple lesions. Of the subjects with multiple lesions, 7.2% had unilateral lesions, 18.5% had bilateral lesions, and 12.4% had lesions in either brainstem or cerebellum. Lesion location was considered unilateral if the lesions were present in only the left or the right hemisphere. Lesion location in the brainstem or cerebellum was indicated as other. Additionally, lesions were classified as cortical, subcortical, and other.

Figure 1 is a visualization of the lesion overlap maps across all subjects (*n* = 655) in the MNI space. 57.1% of subjects had at least one left hemisphere lesion, and 58.8% had at least one right hemisphere lesion [2].

### 2.5. Model Architecture

We applied 3D-UNet, a deep neural network (DNN) architecture, to segment the volumetric imaging data. Our implementation of the 3D-UNet on the ATLAS v2.0 dataset is similar to that of Isensee et al. in the nnUnet framework [22]. Previously, we adapted this architecture to successfully train an automatic segmentation model on the ATLAS v.1.2 dataset [23].

An overview of the 3D-UNet model and its components is shown in Figure 2. The 3D-UNet architecture consists of a symmetrical encoder and decoder structure. During the analysis phase (encoding), convolutions and down sampling operations combine and create a bottleneck at the center of the structure. During the synthesis phase (decoding), a series of convolutions and upsampling (inverse convolutions) reconstructs the image. The addition of skip connections improves training by assisting backward gradient flow [18]. A skip connection allows access to low- and mid-level visual representations of the input to the decoder component, similar to the bypass connections in the primate visual system [24].

### 2.6. Model Training

We optimized the hyperparameters for the training using a grid search and cross-validation on a validation set. Model performance on the validation set determined the hyper-parameters. The model was trained for 1000 epochs (where an epoch is defined as 250 batches). Adaptive learning rates were used during the training. For the first 500 epochs, a cosine-annealing learning rate schedule of 1 × 10^−2^ to 1 × 10^−4^ was used; for epochs 500–1000, a cosine-annealing learning rate schedule of 1 × 10^−5^ to 1 × 10^−9^ was utilized. A weight decay (3 × 10^−5^) method was used for regularization. Scaling, rotation, gamma adjustments, mirror transformations, and additive brightness were randomly performed at probabilities of 0.2, 0.2, 0.3, 0.5, and 0.15, respectively, to augment the 3D data.

### 2.7. Model Performance Evaluation

We assessed the model’s performance both qualitatively and quantitively. For quantitative evaluation, we employed a multitude of evaluation metrics. To compare the spatial overlap of the manual and automatic segmentation labels, we utilized the Sørenson–Dice similarity coefficient (DSC) [25]. It is calculated as:(1)$$DSC=\frac{2\left|R\;\cap^{\;}M\right|}{\left|R\right|+\left|M\right|}\times 100$$
where R is the ground truth reference brain mask (manual segmentation) and M is the predicted mask (automatic segmentation). The output values for DSC range from 0 (no overlap) to 1 (perfect agreement). We complemented DSC with average symmetric surface distance (ASSD). ASSD is the average of all distances between the predicted mask and the ground truth boundary [26]. In addition, we compared lesion volumes across cross-validation folds.

### 2.8. Statistical Analysis

Bootstrap resampling (*n* = 1000) was used to estimate 95% confidence intervals for the evaluation metrics (DSC and ASSD). We report descriptive statistics (median, 95% CI) for DSC and ASSD across cross-validation folds. A two-tailed Wilcoxon matched-pairs signed-rank test was done to test the statistical significance of the differences between the manual and automatic segmentation volumes. For quantification of the relationship between manual and automatic segmentation volumes (randomly selected cross-validation folds, *n* = 2), the Pearson correlation coefficient was utilized. We employed a randomization test to estimate the *p*-value of the correlation coefficient (*n* = 1000).

## 3. Results

We evaluated the performance of our model by comparing manual and automatic segmentation masks across cross-validation folds.

### 3.1. Qualitative Performance

Visual evaluation of performance was done across multiple lesion locations and sizes. To illustrate the model’s optimal performance on both large and small lesions, refer to Figure 3. The images show manual and automatic segmentations of the T1w input image to the model. Across different brain locations, the model can segment lesions of different sizes accurately. Additionally, we provide supplemental videos demonstrating segmentation examples for multiple subjects with different sizes and locations of lesions (see Supplementary Materials).

Figure 4, Figure 5 and Figure 6 illustrate the model’s performance across the full extent of the lesion volume on the coronal, axial, and sagittal sections respectively.

Since Figure 3 shows only one brain slice, Figure 4, Figure 5 and Figure 6 illustrate the model’s performance across the full extent of the lesion volume on the coronal, axial, and sagittal sections. The model provides continuous segmentation solutions across slices and sectional views. As only the model output is displayed in Figure 4, Figure 5 and Figure 6, these qualitative visualizations demonstrate how our model can provide non-degenerate segmentation solutions.

### 3.2. Quantitative Performance

Quantitative performance was evaluated by using DSC, ASSD, and total volume comparisons. These discrepancy metrics assess the similarity and dissimilarity of automatically segmented lesions as compared to the ground truth, that is, the manually segmented label [27]. A summary of descriptive statistics for DSC and ASSD is presented in Table 2.

To demonstrate the accuracy of volume prediction, we measured both manual and automatic segmentation volumes. Scatter plots indicate the linear relationship between manual and automatic segmentations over varying lesion volumes without bias, as most data points lie on the diagonal (Figure 7, data from randomly selected cross-validation folds (*n* = 2)). The Pearson correlation coefficient (ρ) between the two volumes was 0.97 (*p*-value < 0.001, randomization test (*n* = 1000)). A histogram of differences was constructed to complement these results, which showed no significant difference between manual and automatic segmentation volumes (*p*-value = 0.84). The results establish the state-of-the-art performance of 3D-UNet both qualitatively (Figure 3, Figure 4, Figure 5 and Figure 6) and quantitatively (Table 2 and Figure 7). In addition, Figure 7 also shows the stability of segmentation across lesions of varying sizes.

## 4. Discussion

Understanding the brain structure-function relationship remains the fundamental challenge for both clinical and basic neurosciences. Brain lesions such as strokes offer a unique insight into brain function [1]. Although stroke lesions tend to be focal in nature, they usually follow vascular territories and involve concomitant brain areas [28]. Furthermore, the immense variations in stroke lesions and the subsequent functional recovery add to the challenge of delineating the structure-function relationship [10]. Therefore, to study this relationship, one needs a large neuroimaging dataset [29] in combination with the ability to accurately locate lesions.

In this study, we developed an automatic segmentation model using state-of-the-art deep learning techniques and a novel combination of regularization methods. Figure 8 below illustrates a flow chart of the process involved in assessing a chronic stroke lesion quantitatively.

### 4.1. Regularization Methods to Overcome Overfitting

The ultimate test of a prediction model is its generalizability to novel datasets [30]. It is, therefore, essential to avoid overfitting the model on the data it is trained on. Overfitting refers to the model’s high performance on the data it is trained on and a discordant performance on novel data [11]. It is hypothesized that overfitting occurs when the model learns irrelevant aspects of data [11].To address this issue, we used two regularization measures. The first measure was k-fold cross-validation. In k-fold cross-validation, the data is randomly partitioned into k number of subsets, then the model is trained on k−1 subsets (termed as training sets) and tested on the left-out subset (termed as testing set). Another regularization method we utilized is data augmentation [31]. Like k-fold cross validation, this method does not require a novel dataset to test; rather, it capitalizes on the available data. In this method, synthetic distortions such as geometric transformation, rotation, cropping, luminance variation, and magnification are introduced to the images [31]. The goal is to augment the available dataset with the generation of synthetic images [31].

### 4.2. A Combination Approach for Quantitative Analysis

Individual metrics are sensitive to specific aspects of segmentation performance; hence, they are sensitive to different kinds of errors. Therefore, if used singly, they risk ranking an algorithm high that only performs well on a particular aspect. As no single metric is considered the gold standard, a combination approach is recommended. The various kinds of errors that can occur in segmentation tasks include errors in lesion shape, location, and size [26]. Therefore, we utilized a metric targeting each of these errors and reported these under the section quantitative assessment of model performance. 

As the name suggests, overlap metrics are best at capturing the degree of overlapping voxels in the two segmentations; however, they are not as sensitive in detecting discrepancies in the overall spatial distribution of the segmentation [26]. The overlap metric that we used is the Sørenson–Dice similarity coefficient (DSC). This index was originally described by Dice for ecological studies [25], but it has been commonly used in medical image segmentation [32].

To capture location-based errors, the boundary-based-distance method is utilized. These methods assess the dissimilarity of the automatic segmentation with ground truth based on the contour distances of the two lesion masks. The boundary-based-distance metric we used is the average symmetric surface distance (ASSD). ASSD focuses on location errors, but it is not as sensitive in identifying errors in the absolute size [26]. To capture errors in size, we compared the two segmentation volumes. The total volume comparison compares the total sizes, but it does not consider the shape and location of the two lesion masks. Therefore, by employing a combination approach, we ensured a rigorous and comprehensive performance evaluation [26].

### 4.3. Limitations

With any machine learning algorithm, one of the limitations is that it can learn biases from the training dataset. Our algorithm was optimized to be sensitive to manual segmentation labels. Therefore, it was prone to learning any biases associated with the manual segmentation process [33]. Although the dataset was derived from multiple sites, the manual segmentation was done by a common group of tracers, which can introduce subjectivity and confounds [2].To overcome this, the model will need to be tested on independent datasets. This, however, represents another challenge as there are limited publicly available datasets for chronic stroke with manual segmentation labels [9]. Moreover, since the testing is done on a subset of data, it provides evidence for adequate internal validity, however, this provides little information regarding external validity. As this issue is common for automatic segmentation of chronic lesions, the authors of ATLAS v2.0 have created a subset of test data (*n* = 300), which has only preprocessed images available on request. The research groups who develop models are encouraged to request access to this dataset, run the model and submit the automatic labels. In this manner, an objective and unbiased test of generalization can be done [2].

### 4.4. Future Directions

Automatic segmentation models have the potential to serve as an aid in clinical decision-making pertaining to neurorehabilitation [34]. This is a prospective area of personalized medicine wherein accurate predictions regarding patient recovery and relevant rehabilitation programs for individual patients will be made [34]. Furthermore, as strokes tend to reoccur in the same patient, even in cases of acute stroke, the chronic lesion segmentation models can aid in foreseeing the prognosis. Chronic stroke segmentation can predict the pre-stroke modified Rankin score (pre-mRS) [34], which is an independent predictor of poststroke prognosis [35]. This highlights the role of chronic stroke segmentation models in both acute and chronic stroke settings.

From the research standpoint, the combination of this technique with voxel-based lesion-symptom mapping (VLSM) will enable the establishment of new relationships between brain structure and function. VLSM is a lesion-defined method in which the patients are grouped based on similar lesions, and then a comparison regarding behavioral deficit is made on a voxel-by-voxel basis [36,37]. In terms of cognitive function localization, VLSM studies in stroke patients have presented localization data pertaining to deficits in cognition [38], language [39], memory [40], and executive functions [41].

Lastly, we shared our code, detailed model architecture, and trained models in the Supplementary Materials. Data sharing is of special value when it comes to neuroimaging segmentation modeling [42]. In addition to potential future use by other authors, sharing this data enables independent verification, and reproduction of the results, thereby adding to the credibility of our scientific enquiry and method [43].

## 5. Conclusions

For the first time, we report an automated lesion segmentation model on the ATLAS v2.0 dataset. This model facilitates a high-throughput, objective, and automated assessment of chronic stroke lesions. We utilized a novel combination of regularization methods to avoid overfitting. To verify the robustness of our model, we utilized a combination of quantitative metrics. This model, in combination with the latest image analysis techniques, such as VLSM, will enable the study of structure-function relationships. Ultimately, this work will facilitate large-scale neuroimaging analyses for stroke rehabilitation research.

## Figures and Tables

**Figure 1 diagnostics-12-02055-f001:**
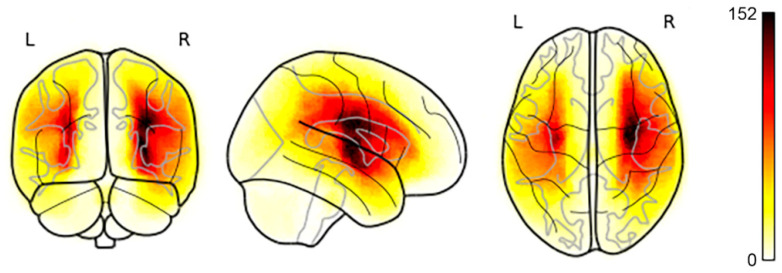
Lesion map across all subjects (*n* = 655) visualized in the standard MNI template. Darker colors indicate a higher frequency of lesions at a given voxel.

**Figure 2 diagnostics-12-02055-f002:**
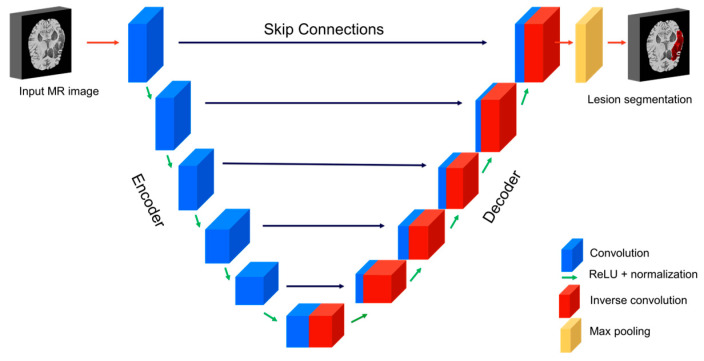
Schematic representation of 3D-UNet architecture.

**Figure 3 diagnostics-12-02055-f003:**
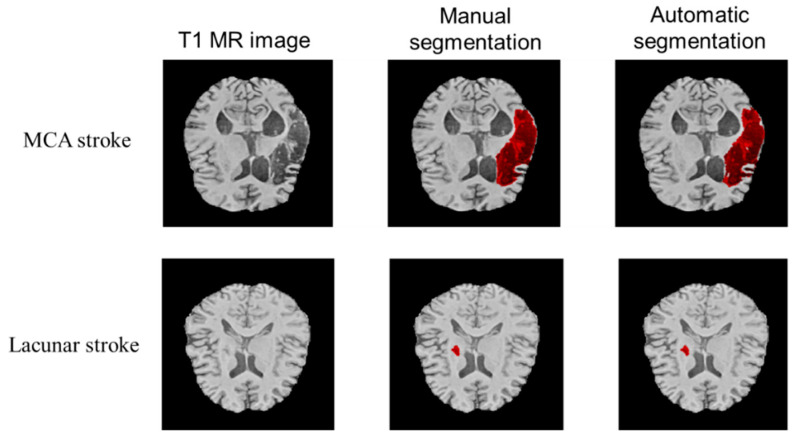
Axial sections showing examples of manual and automatic segmentation. The top row depicts the segmentation performance for a Middle Cerebral Artery (MCA) stroke and the bottom panel demonstrates performance for a lacunar stroke.

**Figure 4 diagnostics-12-02055-f004:**
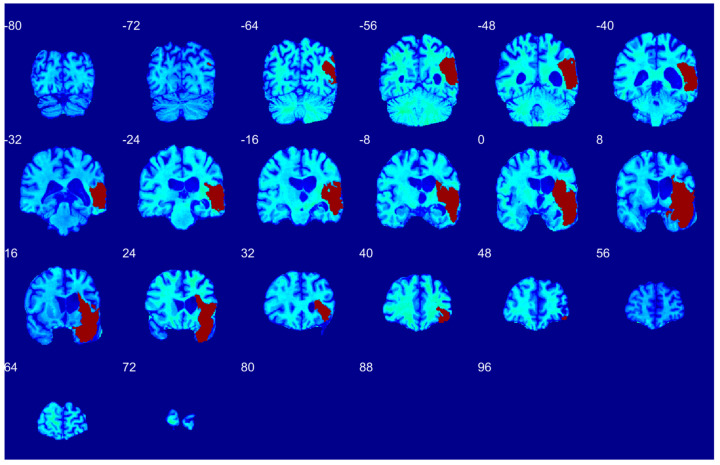
Coronal sections showing automatic segmentation of left MCA stroke as mapped on MNI152 template. The brain slices are sampled at the intervals of 8 mm.

**Figure 5 diagnostics-12-02055-f005:**
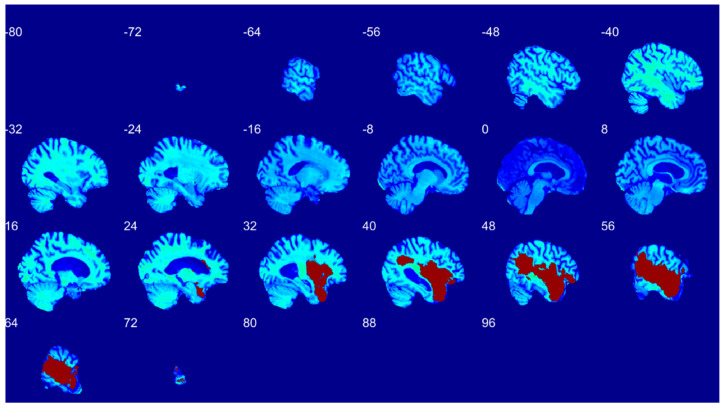
Sagittal sections showing automatic segmentation of left MCA stroke.

**Figure 6 diagnostics-12-02055-f006:**
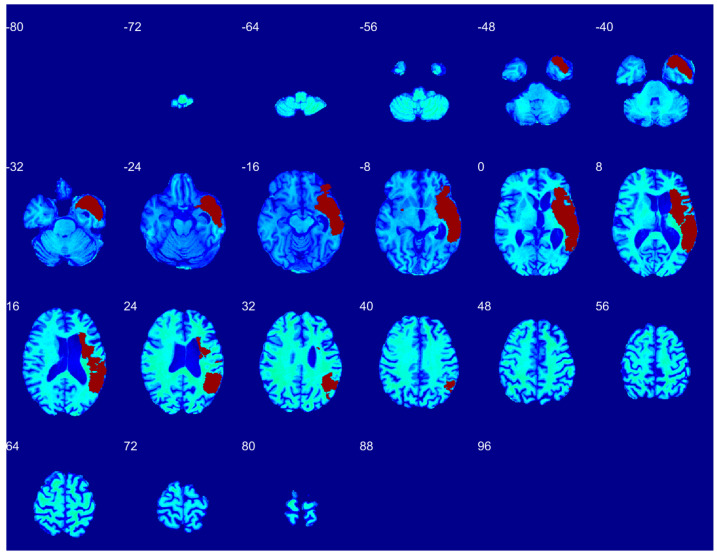
Axial sections showing automatic segmentation of left MCA stroke.

**Figure 7 diagnostics-12-02055-f007:**
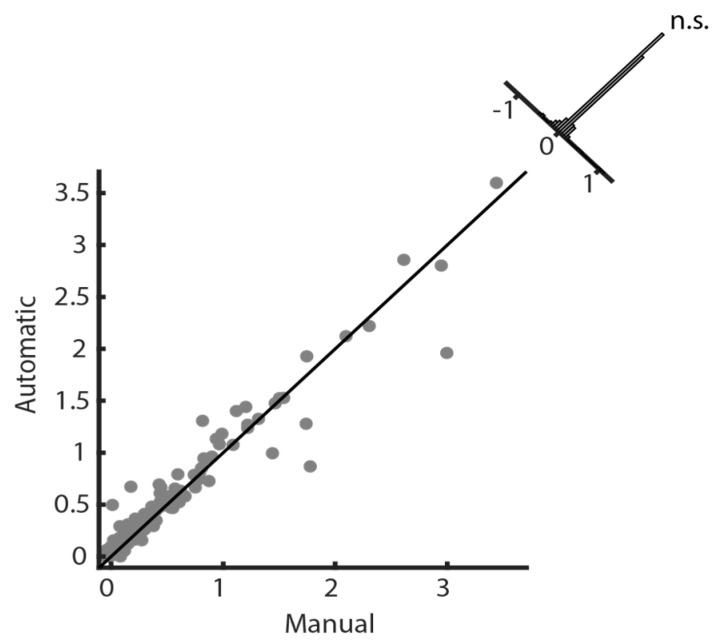
A scatter plot with a histogram of differences shows no significant difference (Wilcoxon rank-sum test, *p-value* = 0.84, n.s. stands for non-significant) between the manual and automatic lesion volumes across cross-validation folds. Segmented lesion volumes (×10^5^ mm^3^). The gray dots represent total lesion volumes for individual subjects from the selected cross-validation folds.

**Figure 8 diagnostics-12-02055-f008:**
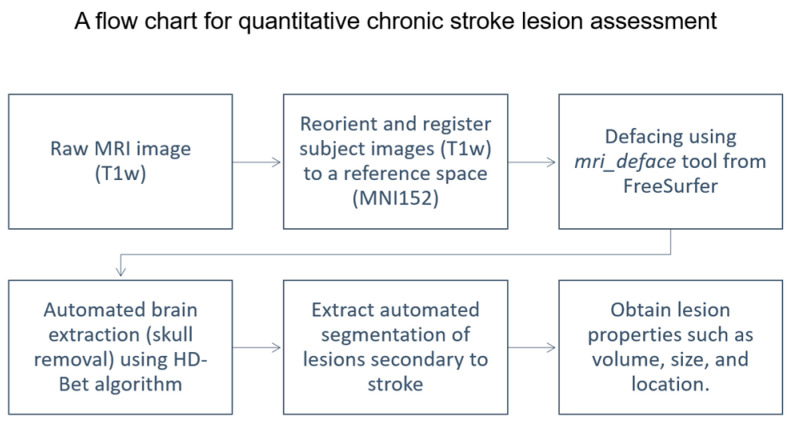
Flow chart detailing the quantitative chronic stroke lesion assessment.

**Table 1 diagnostics-12-02055-t001:** Data characteristics for training dataset.

Subjects (*n* = 655)	Location
Cortical lesions	Left 12%; Right 13.5%
Subcortical lesions	Left 30.2%; Right 29.4%
Other lesions	14.8%

**Table 2 diagnostics-12-02055-t002:** Segmentation performance of the trained model across cross-validated folds.

Performance Metric	Mean (Bootstrap 95% CI)
Sørensen–Dice coefficient	0.65 (0.61–0.67), median: 0.73
Average symmetric surface distance (ASSD) on MNI152 template (in mm)	12.04 (8.44–19.53), median: 2.23

## Data Availability

The link to complete code and pre-trained models is shared in the Supplementary Material.

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
