# Peer review of "Automatic Segmentation and Quantitative Assessment of Stroke Lesions on MR Images"

_diagnostics, 2022, doi:10.3390/diagnostics12092055_

Round 1

Reviewer 1 Report

This is a well-written paper containing interesting results which merit publication.This article is very innovative and presents an MRI algorithm that can be used to automatically segment lesions.For the benefit of the reader,however,a few minor revisions and list below:

(1)In the introduction part,authors shoud give more details about 3D-Unet.

(2)In section 3.1,the author needs to further explain the results of the qualitative analysis and explain how to judge the performance of the model from figures.

(3)Can the quality of figures 4 and 5 be improved again?Does it need to be compared with manual segmentation of lesions.

Overall,the artical is well organized and its presentation is good.However,some minor issues still need to be improved.

Author Response

Our sincere thanks go out to the reviewer who took the time to read our manuscript. Below you will find our point-by-point response in italics.

“(1) In the introduction part,the authors should give more details about 3D-Unet”

  • In the introduction, we have added a paragraph describing 3D-Unet (Page 2).

“In this study, we aim to develop a fully automatic algorithm to segment chronic stroke lesions on T1-weighted MR images. To develop our model, we used a deep neural network architecture (DNN). Among the machine learning algorithms, DNN is a class that has become well-established in computer vision tasks such as automatic segmenta-tion [11]. 3D-UNet has been reported to perform well on a wide range of biomedical seg-mentation challenges in recent studies [7,12–17]. As a variation of 2D-UNet, 3D-UNet takes three-dimensional volume inputs as inputs [18]. The encoder and decoder structures are identical to those in the 2D-UNet. The encoder, however, extracts the features using 3D convolution followed by 3D max-pooling, and the decoder reconstructs the annotated images using 3D upsampling [18]. Therefore, we applied 3D-UNet as it is suitable for volumetric segmentation of medical imaging [19].”

(2) In Section 3.1, The author needs to further explain the results of the qualitative analysis and explain how to judge the performance of the model from pictures. 

  • We thank the reviewer for pointing this out. We now include the additional paragraph detailing the qualitative results (page 5) and further supplemental videos that demonstrate segmentation examples for multiple subjects with different sizes and locations of lesions in the supplementary materials.
    “Visual evaluation of performance was done across multiple lesion locations and sizes. To illustrate the model’s optimal performance on both large and small lesions, refer to figure 3. The images show manual and automatic segmentations of the T1w input image to the model to evaluate its performance. Across different brain locations, the model can segment lesions of different sizes accurately. Additionally, we provide supplemental vid-eos that demonstrate segmentation examples for multiple subjects with different sizes and locations of lesions (see Supplementary Materials).”

(3) Can the quality of figures 4 and 5 be improved again? Does it need to be compared with manual segmentation of lesions.

  • We appreciate this suggestion. In addition to the qualitative results, we provide supplementary videos to accompany figures 3, 4, 5, and 6. Additionally, we have added a paragraph describing these figures (page 7).
    “Since Figure 3 shows only one brain slice, Figures 4, 5, and 6 illustrate the model's performance across the full extent of the lesion volume on the coronal, axial, and sagittal sections. The model provides continuous segmentation solutions across slices and sectional views. As only the model output is displayed in Figures 4, 5, and 6, these qualitative visualizations demonstrate how our model is capable of providing non-degenerate segmentation solutions.”

Reviewer 2 Report

This paper studies Automatic Segmentation and Quantitative Assessment of  Stroke Lesions on MR Images. DNN enabled objective and automated assessment of chronic lesions secondary to stroke in vascular neurology at high throughput and could ultimately serve to improve clinical decision-making. Meanwhile, the voxel-based lesion-symptom mapping will facilitate the study of brain-behavior relationships in chronic stroke. This is an interesting paper that could be a potentially publishable subject. Some general and main weaknesses should be addressed in this paper. Therefore, I suggest the authors resubmit it after a Major revision. My suggestions are:

General comments:

1. In subsections 4.3 and 4.4 you have provided the limitations and future studies. Discuss more the limitations of the study and future research suggestions, if any.

2. The paper should be revised to include at least 10 recent (2020-2021-2022) high-quality stroke-relevant references including:

Intelligent Algorithm-Based MRI Image Features for Evaluating the Effect of Nursing on Recovery of the Neurological Function of Patients with Acute Stroke

-Convolutional neural network with batch normalization for glioma and stroke lesion detection using MRI

-Consistent depiction of the acidic ischemic lesion with APT MRI—Dual RF power evaluation of pH-sensitive image in acute stroke

-Artificial intelligence in stroke imaging: Current and future perspectives

-Image fusion practice to improve the ischemic-stroke-lesion detection for efficient clinical decision making

-Developing a Novel Integrated Generalised Data Envelopment Analysis (DEA) to Evaluate Hospitals Providing Stroke Care Services

-Hybrid Image Processing-Based Examination of 2D Brain MRI Slices to Detect Brain Tumor/Stroke Section: A Study

3. Your introduction is too short. Meanwhile, please consider a literature review part in the introduction as well. Please separate the introduction and literature review.

4. The quality of English needs to be improved across the paper. Also, the scientific terms pertinent to the Stroke topic.

Main comments:

1. Since your model is the first automated segmentation model trained on a large multicentric dataset and it will enable automated on-demand processing of MRI scans as well as quantitative chronic stroke lesion assessment a flowchart is beneficial, it’s also important to outline the methodology behind your suggested approach. Please consider a flowchart of the suggested approach in the methodology part of your paper.

2. Some of your subsections for material and methods are too short. You have 8 subsections which are too short. Please explain more. In particular Preprocessing and Dataset Partitioning as well as Statistical Analysis.

3. In lines 157-159 you have mentioned: "Visual evaluation of performance was done across multiple lesion locations and sizes. To illustrate the model’s optimal performance on both large and small lesions, refer to figure 3" Please explain more about Qualitative Performance. In fact, functional performance enrichment analysis is so important and this part needs more clarification. Only a short explanation and referring to a reference is not enough in this part and all other parts.

4. In lines 164-165 you have provided a short explanation of figures 4-6 which is "Figures 4, 5, and 6 illustrate the model’s performance across the full extent of the lesion volume on the coronal, axial, and sagittal sections respectively"

Please add at least one paragraph and discuss the results regarding the outputs of the suggested figures.

5. In lines 181-183 you have mentioned: "To demonstrate accurate volume prediction, a scatter plot with a histogram of differences was constructed which showed no significant difference (p-value = 0.84) between the manual and automatic segmentation volumes" Please explain more about your foundation in scatter plot. It is so important to bold and clarifies your foundations.

6. In lines 237-238 you have mentioned "Total volume comparison is best for comparing the total sizes"

Why? How can you prove your theory? Please explain and discuss more. 

Author Response

Our sincere thanks go out to the reviewer who took the time to read our manuscript. Below you will find our point-by-point response in italics.

General comments:

  1. In subsections 4.3 and 4.4 you have provided the limitations and future studies. Discuss more the limitations of the study and future research suggestions, if any.
  • We thank the reviewer for this suggestion. We have updated sections 4.3 and 4.4 to incorporate the suggestions (page 10).

4.3. Limitations

With any machine learning algorithm, one of the limitations is that it can learn biases from the training dataset. Our algorithm was optimized to be sensitive to manual segmentation labels, therefore it was prone to learn any biases associated with the manual segmentation process [34]. Although the dataset was derived from multiple sites, yet the manual segmentation was done by a common group of tracers, which can introduce subjectivity and confounds [35].To overcome this, the model will need to be tested on independent datasets. This, however, represents another challenge as there are limited publicly available datasets for chronic stroke with manual segmentation labels [9]. Moreover, since the testing is done on a subset of data, it provides evidence for adequate internal validity, however, provides little information regarding external validity. As this issue is common for automatic segmentation of chronic lesions, the authors of ATLASv2.0 have created a subset of test data (n =300), which has only preprocessed images available to request. The research groups who develop models are encouraged to request access to this dataset, run the model and submit the automatic labels. In this manner, an objective and unbiased test of generalization can be done [35].

4.4. Future Directions

Automatic segmentation models have the potential to serve as an aid in clinical decision-making pertaining to neurorehabilitation [36]. This is a prospective area of personalized medicine wherein more accurate predictions regarding patient’s recovery and relevant rehabilitation programs for individual patients will be made [36]. Furthermore, as strokes tend to reoccur in the same patient, even in cases of acute stroke, the chronic lesion segmentation models can aid in foreseeing the prognosis. Chronic stroke segmentation can predict the pre-stroke modified rankin score (pre-mRS) [36], which is an independent predictor of poststroke prognosis [37]. This highlights the role of chronic stroke segmentation models in both acute and chronic stroke settings.

From the research standpoint, the combination of this technique with voxel-based lesion-symptom mapping (VLSM) will enable the establishment of new relationships between brain structure and function. VLSM is a lesion-defined method in which the patients are grouped on the basis of similar lesions and then comparison regarding behavioral deficit is made on voxel-by-voxel basis[38,39]. In terms of cognitive function localization, VLSM studies in stroke patients have presented localization data pertaining to deficits in cognition[40], language[41], memory[42], and executive functions.[43]

Lastly, we shared our code, detailed model architecture, and trained models in the supplementary materials. Data sharing is of special value when it comes to neuroimaging segmentation modeling [44]. In addition to potential future use by other authors, sharing this data enables independent verification, and reproduction of the results, thereby adding to the credibility of our scientific enquiry and method [45]. “

  1. The paper should be revised to include at least 10 recent (2020-2021-2022) high-quality stroke-relevant references including:
  • We have now included additional up-to-date, relevant, and high-quality references. We cannot include some of the references suggested by the reviewer because they were irrelevant to our work or of poor methodological quality. We include the following additional references:
    • Tomita, N.; Jiang, S.; Maeder, M.E.; Hassanpour, S. Automatic Post-Stroke Lesion Segmentation on MR Images Using 3D Residual Convolutional Neural Network. NeuroImage Clin. 2020, 27, 102276, doi:https://doi.org/10.1016/j.nicl.2020.102276.
    • Paing, M.P.; Tungjitkusolmun, S.; Bui, T.H.; Visitsattapongse, S.; Pintavirooj, C. Automated Segmentation of Infarct Lesions in T1-Weighted MRI Scans Using Variational Mode Decomposition and Deep Learning. Sensors 2021, Vol. 21, Page 1952 2021, 21, 1952, doi:10.3390/S21061952.
    • Xue, Y.; Farhat, F.G.; Boukrina, O.; Barrett, A.M.; Binder, J.R.; Roshan, U.W.; Graves, W.W. A Multi-Path 2.5 Dimensional Convolutional Neural Network System for Segmenting Stroke Lesions in Brain MRI Images. NeuroImage Clin. 2020, 25, 102118, doi:10.1016/J.NICL.2019.102118.
    • Chen, X.; You, S.; Tezcan, K.C.; Konukoglu, E. Unsupervised Lesion Detection via Image Restoration with a Normative Prior. Med. Image Anal. 2020, 64, 101713, doi:10.1016/J.MEDIA.2020.101713.
    • Hui, H.; Zhang, X.; Li, F.; Mei, X.; Guo, Y. A Partitioning-Stacking Prediction Fusion Network Based on an Improved Attention U-Net for Stroke Lesion Segmentation. IEEE Access 2020, 8, 47419–47432, doi:10.1109/ACCESS.2020.2977946.
    • Liu, X.; Yang, H.; Qi, K.; Dong, P.; Liu, Q.; Liu, X.; Wang, R.; Wang, S. MSDF-Net: Multi-Scale Deep Fusion Network for Stroke Lesion Segmentation. IEEE Access 2019, 7, 178486–178495, doi:10.1109/ACCESS.2019.2958384.
    • Zhang, Y.; Wu, J.; Liu, Y.; Chen, Y.; Wu, E.X.; Tang, X. MI-UNet: Multi-Inputs UNet Incorporating Brain Parcellation for Stroke Lesion Segmentation from T1-Weighted Magnetic Resonance Images. IEEE J. Biomed. Heal. Informatics 2021, 25, 526–535, doi:10.1109/JBHI.2020.2996783.
    • Çiçek, Ö.; Abdulkadir, A.; Lienkamp, S.S.; Brox, T.; Ronneberger, O. 3D U-Net: Learning Dense Volumetric Segmentation from Sparse Annotation. Lect. Notes Comput. Sci. (including Subser. Lect. Notes Artif. Intell. Lect. Notes Bioinformatics) 2016, 9901 LNCS, 424–432, doi:10.1007/978-3-319-46723-8_49.
    • Zhou, Y.; Huang, W.; Dong, P.; Xia, Y.; Wang, S. D-UNet: A Dimension-Fusion U Shape Network for Chronic Stroke Lesion Segmentation. IEEE/ACM Trans. Comput. Biol. Bioinforma. 2021, 18, 940–950, doi:10.1109/TCBB.2019.2939522.

  1. Your introduction is too short. Meanwhile, please consider a literature review part in the introduction as well. Please separate the introduction and literature review.
  • We thank the reviewer for the suggestion. We have now included an additional paragraph citing relevant literature and approaches (page 2). It is, however, beyond the scope of this research to conduct a detailed comparative literature review.

“In this study, we aim to develop a fully automatic algorithm to segment chronic stroke lesions on T1-weighted MR images. To develop our model, we used a deep neural network architecture (DNN). Among the machine learning algorithms, DNN is a class that has become well-established in computer vision tasks such as automatic segmentation [11]. Recent studies have reported that DNNs, specifically 3D-UNet, are effective at segmenting stroke lesions [7,12–17]. As a variation of 2D-UNet, 3D-UNet takes three-dimensional volume inputs as inputs [18]. The encoder and decoder structures are identical to those in the 2D-UNet. The encoder, however, extracts the features using 3D convolution followed by 3D max-pooling, and the decoder reconstructs the annotated images using 3D upsampling [18]. Therefore, we applied 3D-UNet as it is suitable for volumetric segmentation of medical imaging [19].”

  1. The quality of English needs to be improved across the paper. Also, the scientific terms pertinent to the Stroke topic.
  • The manuscript was proofread by several native speakers, including Christina Roth, an experienced professional editor. Please see the acknowledgments for more information. 

Main comments:

  1. Since your model is the first automated segmentation model trained on a large multicentric dataset and it will enable automated on-demand processing of MRI scans as well as quantitative chronic stroke lesion assessment a flowchart is beneficial, it’s also important to outline the methodology behind your suggested approach. Please consider a flowchart of the suggested approach in the methodology part of your paper.
  • An additional figure has been added. A flow chart of the process involved in assessing a chronic stroke lesion quantitatively is illustrated in Figure 8.
  1. Some of your subsections for material and methods are too short. You have 8 subsections which are too short. Please explain more. In particular Preprocessing and Dataset Partitioning as well as Statistical Analysis.
  • To address the concern raised by the reviewer we have rewritten the subsection preprocessing and Dataset Partitioning including additional details (page 3). 

“Preprocessing included intensity normalization, registration to a standardized template (MNI-152), and defacing using the “mri_deface” tool from FreeSurfer (v1.22) (https://surfer.nmr.mgh.harvard.edu/fswiki/mri_deface). Automated brain extraction using the HD-BET algorithm was performed on all the MR images [22]. HD-BET is a rigorously validated deep learning algorithm, which uses artificial neural networks. During our initial testing, we found that the brain extraction (skull removal) process reduces the training time and improves the model performance. For training, the dataset was partitioned into unique five-folds containing 20% of the data in each fold.”

  1. In lines 157-159 you have mentioned: "Visual evaluation of performance was done across multiple lesion locations and sizes. To illustrate the model’s optimal performance on both large and small lesions, refer to figure 3" Please explain more about Qualitative Performance. In fact, functional performance enrichment analysis is so important and this part needs more clarification. Only a short explanation and referring to a reference is not enough in this part and all other parts.
  • We thank the reviewer for this suggestion. We now include the additional paragraph detailing the qualitative results (page 5) and further supplemental videos that demonstrate segmentation examples for multiple subjects with different sizes and locations of lesions in the supplementary materials.

    “Visual evaluation of performance was done across multiple lesion locations and sizes. To illustrate the model’s optimal performance on both large and small lesions, refer to figure 3. The images show manual and automatic segmentations of the T1w input image to the model to evaluate its performance. Across different brain locations, the model can segment lesions of different sizes accurately. Additionally, we provide supplemental vid-eos that demonstrate segmentation examples for multiple subjects with different sizes and locations of lesions (see Supplementary Materials).”
  1. In lines 164-165 you have provided a short explanation of figures 4-6 which is "Figures 4, 5, and 6 illustrate the model’s performance across the full extent of the lesion volume on the coronal, axial, and sagittal sections respectively" Please add at least one paragraph and discuss the results regarding the outputs of the suggested figures.
  • As mentioned in the previous point, we have now added additional details for the figures and supplemental videos to accompany the results (see Supplementary Materials). A paragraph has also been added to explain and improve the interpretation of Figures 4, 5, and 6 (page 7).

    “Since Figure 3 shows only one brain slice, Figures 4, 5, and 6 illustrate the model's performance across the full extent of the lesion volume on the coronal, axial, and sagittal sections. The model provides continuous segmentation solutions across slices and sectional views. As only the model output is displayed in Figures 4, 5, and 6, these qualitative visualizations demonstrate how our model is capable of providing non-degenerate segmentation solutions.”

  1. In lines 181-183 you have mentioned: "To demonstrate accurate volume prediction, a scatter plot with a histogram of differences was constructed which showed no significant difference (p-value = 0.84) between the manual and automatic segmentation volumes" Please explain more about your foundation in scatter plot. It is so important to bold and clarifies your foundations.
  • To address this concern by the reviewer, we have updated this section (page 8).

“In order to demonstrate the accuracy of volume prediction, we measured both manual and automatic segmentation volumes. Scatter plots indicate the linear relationship between manual and automatic segmentations over varying lesion volumes, without bias, as most data points lie on the diagonal. The Pearson correlation coefficient (ρ) between the two volumes was 0.97(p-value < 0.001, randomization test (n = 1000)). A histogram of differences was constructed to complement these results, which showed no significant difference between manual and automatic segmentation volumes (p-value = 0.84). The results establish the state-of-the-art performance of 3D-UNet both qualitatively (Figures 3-6) and quantitatively (Table 2 and Figure 7). In addition, Figure 7 also shows the stability of segmentation across lesions of varying sizes.”

  •  
  1. In lines 237-238 you have mentioned "Total volume comparison is best for comparing the total sizes"

Why? How can you prove your theory? Please explain and discuss more. 

  • We have now updated the sentence and included references to the conceptual theory (page 9).

    The total volume comparison compares the total sizes, but it does not consider the shape and location of the 2 lesion masks. Therefore, by employing a combination approach we ensured a rigorous and comprehensive performance evaluation [27]. “

Round 2

Reviewer 2 Report

The authors just answered all my comments except one main comment regarding the introduction part. Please mention the structure of your paper at the end of the introduction. For example:

The remainder of the paper is organized as follows. Section 2 presents the background of the research. In Section 3........ Section 4 provides a brief definition of the........ In Section 5......

Author Response

At the end of the introduction, we have included the additional part mentioning the paper's structure as suggested by the reviewer.

"The rest of the paper follows the following structure. We begin by describing how the model was developed. A qualitative and quantitative evaluation of the model will follow in the next section. We conclude by discussing the application of our developed model in both basic and clinical research."